# Development of On-Demand Antiviral Electrostatic Precipitators with Electrothermal-Based Antiviral Surfaces against Airborne Virus Particles

**DOI:** 10.3390/toxics10100601

**Published:** 2022-10-11

**Authors:** Dae Hoon Park, So-Hee An, Yeawan Lee, Yong-Jin Kim, Bangwoo Han, Hak-Joon Kim

**Affiliations:** 1Department of Sustainable Environment Research, Korea Institute of Machinery and Materials, Daejeon 34103, Korea; 2Department of Mechanical Engineering, Yonsei University, Seoul 03722, Korea

**Keywords:** electrostatic precipitator, electrothermal effect, on-demand process, antiviral, airborne virus

## Abstract

Particulate matter, including airborne pathogens, is of particular concern because it can cause the spread of diseases through aerosol transmission. In this study, a new concept is proposed: on-demand antiviral electrostatic precipitators (ESPs) with electrothermal-based antiviral surfaces. We applied electrothermal-based antiviral surfaces to air-purifying applications and demonstrated that the proposed method is effective with regard to collecting airborne virus particles on collection plates in a two-stage ESP. With alternating current power, MS2 bacteriophage and H1N1 viruses were completely deactivated after exposure to 50 °C for 30 min. This remarkable antiviral performance via electrothermal effects indicates that on-demand platforms for self-antiviral surfaces can perform sterilization immediately without generating secondary pollutants, thus effectively preventing the spread of infectious microorganisms in public places. We believe that the results of this study can provide useful guidelines for the design and realization of practical and wearable devices for antiviral air-purifying applications.

## 1. Introduction

Global concerns over air contamination have continuously risen over the past decades. Air pollutants can be risky for many individuals who live in confined places and do not have sufficient ventilation, especially for people with weak immune systems such as children, seniors, and hospital patients [1,2,3,4]. The components of air contaminants vary and can contain toxic chemicals and particulate matter (PM), such as airborne pathogens. Especially very recently, airborne viruses are of particular concern because they can cause the spread of diseases through aerosol transmission. The COVID-19 pandemic, a worldwide outbreak of the severe acute respiratory syndrome coronavirus 2 (SARS-CoV-2), is a representative example. According to the U.S. Centers for Disease Control and Prevention (CDC), SARS-CoV-2 is transmitted through exposure to infectious respiratory fluids caused by infected individuals through talking, coughing, or sneezing, thus spreading virions in airborne droplets and aerosol forms [5,6,7]. Thus, tremendous research efforts have focused on the development of efficient strategies for the removal of airborne viral pathogens [8,9,10,11].

To mitigate the exposure of airborne viral particles in indoor air conditions, air filters or electrostatic precipitators (ESPs) are usually used to capture viral aerosols on the surfaces of filter fibers or collection plates in air-purifying systems. However, previous studies have reported that viral particles can remain active on various surfaces, such as plastic, paper, and stainless steel, for significantly longer time periods than generally expected (over a few days) [12,13,14], which prompted considerable efforts on sterilizing all surfaces so as to suppress the potential risks of fomite-mediated infections [15]. Many types of antiviral technologies, which were fabricated by coating the surfaces where viral particles are captured with functional materials, have been suggested in recent reports [16,17,18,19,20]. However, a challenge for these coating-based antiviral strategies is the renewal of antiviral activities for long-term care because of the performance deterioration of coating materials and the accumulation of dust particles over the coating surfaces [18,19].

Consequently, developing an on-demand platform for self-sterilizing surfaces may be an important advancement. An on-demand antiviral method with heat treatment for inactivating captured viruses on surfaces has recently been introduced as a promising and alternative antiviral technology [14,21] that can immediately perform sterilization without generating secondary pollutants, thus effectively preventing the spread of infectious microorganisms in public places.

In our previous study, we presented an antimicrobial carbon surface with an electrothermal effect, and it showed outstanding antimicrobial performance against gram-positive and gram-negative bacteria. Moreover, we demonstrated the electrothermal-based antimicrobial mechanism and its optimal conditions for achieving maximum efficiency [14]. In this study, we applied these hybrid electrothermal-based surfaces on a two-stage ESP for an on-demand antiviral system (not on filter). Moreover, the electrothermal-based antiviral performance against airborne pathogens, including infectious viruses (not only bacteria), was tested. Therefore, we herein newly introduce an on-demand antiviral ESP with electrothermal-based antiviral surfaces.

The newly presented strategy for the rapid inactivation of airborne virus particles with functional ESPs would be attractive for lightweight, cost-effective, harmless, and energy-efficient air-purifying applications that can be used to prevent the transmission of infectious viruses. Overall, we believe that this study can potentially be used for the design and realization of practical and wearable antiviral and air-purifying devices.

## 2. Materials and Methods

### 2.1. Electrothermal Surface Preparation

In our previous study, we applied alternating current (AC) power to carbon surfaces with microorganisms, which were completely deactivated after exposure to 50 °C for 10 min [21]. In this study, we selected one of the commercially available carbon surfaces, the performance of which was demonstrated in a previous study [21]. The selected carbon surface had low electric resistance (~5.84 × 10^−^^2^ Ω·m) to achieve high surface temperature with low energy consumption and was coated with a polyethylene terephthalate (PET) layer to realize lightweight and flexible properties [22], thus ensuring an energy-efficient antiviral ESP. Figure 1a shows a digital photo of the carbon surface, which had a width, height, and thickness of 102.1, 21.8, and 0.3 mm, respectively. Figure 1b shows the experimental setup of the electrical resistance test for the characterization of electrothermal properties with AC power using SLIDE-AC. The generated current–voltage, which was measured by a multimeter, was applied to the carbon surface to generate heat, and the surface temperature was measured using a thermocouple.

### 2.2. Preparation of Test Virus Solutions

In this study, MS2 bacteriophage (ATCC 15597-B1) and H1N1 (influenza A/california/07/2009) were used as test virus species. MS2 bacteriophage was purchased from American Type Culture Collection (ATCC), VA, USA. For the preparation of the MS2 bacteriophage virus solution, a frozen virus stock (0.1 mL) was defrosted at room temperature and inoculated into 50 mL of deionized pure water. To measure the bacteriophage concentration of the stock, a plaque assay was performed (initial concentration: 10^8^ PFU mL^−1^). H1N1 was provided by the BioNano Health Guard Research Center, Daejeon, Korea (initial concentration: 2.8 × 10^7^ PFU mL^−1^).

### 2.3. Characterization of the Functional ESP Performance

In this study, a two-stage ESP, which consisted of a charging part and a collection part, was developed for the PM removal test with electrothermal carbon surfaces. A detailed view of the two-stage ESP is shown in Figure 2. The charging part (40 × 40 × 30 mm^3^) consisted of a carbon brush-type ionizer located at the center of the duct (Figure 2, left), and the collection part (40 × 40 × 100 mm^3^) consisted of collection plates and the ground plates with a 2-mm gap distance (Figure 2, right). In the charging part, a commercial carbon brush-type ionizer was selected with an average fiber diameter of ~7 μm, which is optimal for discharge electrodes to minimize ozone generation [22,23,24,25] with negative high voltage. Moreover, a plastic (polypropylene; PP) coating was applied to the metal ground electrode of the discharging stage to effectively suppress the generation of ozone [22] in the charging part of the two-stage ESP. In the collection part, PET-coated carbon surfaces were used for both the collection plate and ground electrodes. With the flexibility of the PET coating surfaces, the gap distance between the electrodes could be maintained with a high specific surface area and a light weight [22].

An experimental schematic of the single-pass PM removal efficiency test of the two-stage ESP is shown in Figure 3. Atmospheric dust was selected as the test PM for the performance evaluation of the two-stage ESP. The test duct had an area of 0.04 × 0.04 m, and the flow velocity of the test duct was adjusted to ~2 m/s by controlling the commercial fan speed. The applied voltage to the fiber brush in the ionization stage was controlled from 0 to −6 kV using a negative high-voltage power supply. Due to the relatively sufficient oxygen, the negative ions were more likely to exist in atmospheric conditions (compared with indoor conditions) [26,27]. To obtain high collection efficiency with high charge density from the abundant negative ions, negative high voltage was applied to generate corona discharge. A voltage of −2 kV was also applied to the collection plate. The applied voltages and corona currents were measured using digital multimeters (Model 286; Fluke Corp., Tokyo, Japan), and the voltage was varied from 0 to −6 kV to the ionizer to investigate the corona discharge properties. The PM concentration was measured using an optical particle counter (OPC, Model 1.109; Grimm, Germany) located at the rear end of the collection stage to measure the PM collection efficiency, which was calculated by (1):
(1)$$\eta_{PM}\left(\%\right)=\left(1-\frac{C_{off}}{C_{on}}\right)\times 100,$$
where $\eta_{PM}$ denotes the PM collection efficiency, and $C_{off}$, $C_{on}$ indicate the downstream particle number concentrations from the two-stage ESP when it is OFF and ON, respectively. To evaluate the PM removal performance, a concentration of 0.3 μm particles was applied to Equation (1), as it is difficult to collect 0.3 μm particles due to their low particle charging rate, which is derived from the combined charging mechanism between diffusion and field charging [28].

The ozone concentrations were measured using an ozone monitor (model 400E, Teledyne Technologies Inc., San Diego, CA, USA) (Figure 3). The net ozone concentration generated from the ESP can be calculated by (2):(2)$$O_{3,net}=O_{3,on}-O_{3,off},$$
where $O_{3,net}$ indicates the volume concentrations of the net ozone, and $O_{3,on}$ and $O_{3,off}$ denote the downstream ozone concentrations from the two-stage ESP when it is OFF and ON, respectively.

### 2.4. On-Demand Antiviral Performance Evaluation of the Functional ESP against Airborne Viruses

MS2 bacteriophage and H1N1 viruses were selected as target airborne viruses. The virus solution was aerosolized by an atomizer in 2 L/min of compressed clean air. Aerosolized virus particles entered the test duct through a diffusion dryer in order to remove any moisture, and 3 L/min of clean air was used as sheath air. The flow velocity of the acrylic duct was adjusted to 0.56 m/s, and the applied voltage of −5 kV to the charging part was controlled. A voltage of −2 kV was applied to the collection part using a high-voltage power supply. The concentration of the airborne virus particles was measured using a scanning mobility particle sizer (SMPS) located upstream and downstream of the ESP (Figure 4a). Then, the collection efficiency of the ESP was calculated as follows:
(3)$$\eta_{virus}\left(\%\right)=\left(1-\frac{C_{down}}{C_{up}}\right)\times 100,$$
where $\eta_{virus}$ denotes the virus collection efficiency, and $C_{down}$, $C_{up}$ indicate the particle number concentrations in the downstream and upstream parts of the two-stage ESP, respectively, via SMPS measurement data.

To evaluate the electrothermal antiviral efficacy, after virus collection for 30 min, the collection surface was electrothermally heated to 50 °C for 30 min. Figure 4b shows the concept of the on-demand electrothermal antiviral process in an ESP, showing that the first step is the collection of airborne viruses on the collection plate via the ESP with direct-current (DC) power and that the second step involves electrothermal treatment against collected viruses with AC power via a simple switching circuit (whenever a user wants to sterilize the surfaces).

Finally, after the electrothermal treatment, the collection surfaces were placed in 5 mL of a phosphate-buffered saline (PBS) solution, and vortexed for 15 min to detach viruses from the collection surface into the PBS solution. After a serial dilution, the viable concentrations of the MS2 bacteriophage and H1N1 viruses were evaluated using plaque assays [29] (Figure 4c) and a reverse transcription–polymerase chain reaction (RT-PCR) system (12675885, Thermo Scientific™ PikoReal™, Vantaa, Finland; Figure 4d), respectively [30].

For the MS2 bacteriophages, the antiviral efficiency of the sample was calculated using the following equation:(4)$$\eta_{antiviral,MS2}\left(\%\right)=\left(1-\frac{PFU_{treated}}{PFU_{untreated}}\right)\times 100,$$
where PFU indicates the concentration of the MS2 bacteriophage particles, and the subscripts treated and untreated indicate with and without electrothermal treatment, respectively.

For the H1N1 viruses, the antiviral efficiency of the sample was determined based on the cycle number obtained from the PCR analysis as follows:(5)$$\eta_{antiviral,\;H1N1}\left(\%\right)=\left(1-\frac{1}{2^{C_{treated}-C_{untreated}}}\right)\times 100,$$
where $C_{treated}$ and $C_{untreated}$ represent the cycle numbers for the samples with and without electrothermal treatment, respectively.

## 3. Results and Discussion

### 3.1. Electrothermal Surface Preparation

Figure 5 shows the change in surface temperature and current according to the applied voltage under AC power conditions. With an AC current of 15.1 A, the surface temperature of the plate increased to 50 °C in <1 min, and this temperature was maintained for a long time (over 30 min), indicating that the carbon-based surface can easily sustain temperatures up to 100 °C [31]. In accordance with our previous study, a clear tendency observed was that the temperature was linearly related to the AC current. Thus, the selected carbon surface can be suitable for electrothermal antiviral surfaces, as it was reported that the deactivation of microorganisms was completed after exposure to 50 °C for 10 min and not only with low resistance but also with efficient heat production [21].

### 3.2. Characterization of the Functional ESP Performance

The inset of Figure 6 shows the corona discharge currents of the charging part with the increase in applied voltage. The corona onset voltage was approximately −2.1 kV, and the maximum current magnitude was −0.031 mA at an applied voltage of −6 kV to the charging part. Figure 6 also shows the PM collection efficiencies and ozone concentrations when varying the applied voltage of the charging part against atmospheric dust. The PM collection efficiencies of the ESP increased with the increase in the voltage of the charging part, and the maximum efficiency was ~80% with an applied voltage of −6kV to the charging part. The ozone concentration was only 14.6 ppb when the collection efficiency was ~75%. In comparison with various regulations of ozone concentrations, such as the World Health Organization (WHO) guidelines (<50 ppb for 8 h) [32], National Ambient Air Quality Standards (NAAQS) guidelines (average 70 ppb for 8 h) [33], and Korea Air Cleaning Association (KACA) regulation (average of 30 ppb for 24 h) [34], the obtained concentration is significantly low. Thus, the proposed functional ESP has significant potential for indoor air-management applications.

### 3.3. On-Demand Aniviral Performance Evaluation of the Functional ESP against Airborne Viruses

Figure 7 shows the results of the airborne virus removal efficiency test of the ESP with voltages of −5 kV and −2 kV applied to the charging part and collection part, respectively. In the cases of all flow velocities, the collection efficiencies against both virus species (MS2 and H1N1) were higher than 90% because the size distributions of the airborne viruses—the mode diameters of which were 60 nm and 80 nm with the MS2 and H1N1 virus particles upstream of the ESP, respectively (Figure 7; right)—have higher electrical mobility than the 0.3 μm of atmospheric dust. The particles with sizes between 0.1 and 0.5 µm had the minimum electrical mobility, and their range corresponded to the most penetrating particle size (MPPS), indicating that this range of particles has less ability to move in an electric field, resulting in a lower collection efficiency in the collection part. However, higher electrical mobility for the particles in the range of 60–100 nm in diameter, whose main charging mechanism is diffusion, resulted in higher collection efficiency in the collection part [35,36]. Moreover, the migration velocities, which are proportional to the Cunningham coefficient, of the 60 nm and 80 nm particles were higher than those of the 0.3 um particles due to the decrease in the fluid drag force [37].

To investigate the electrothermal antiviral efficacy after virus collection for 30 min, the collection surface was electrothermally heated to 50 °C for 30 min. Figure 8 shows the electrothermal antiviral activity of the carbon surface against the MS2 bacteriophage. The results show that the electrothermally heated carbon surface has a tremendous antiviral effect in a short time, as the thermal energy was sufficient for destroying the proteins in the viruses, thus deactivating them [14,29]. After just 30 min, the electrothermal-based antiviral efficiency against the captured MS2 bacteriophage on the carbon surface was over 99.99% (the number of virus concentrations was under the detection of limit (<10 PFU/mL)), indicating that the viruses were completely deactivated.

Figure 9 shows the electrothermal antiviral activity of the carbon surface against H1N1 viruses. These results show a similar tendency against MS2 bacteriophage. However, H1N1 viruses are likely more susceptible to electrothermal antiviral activity.

We demonstrated that the concept of the on-demand electrothermal antiviral process in an ESP is feasible with remarkable antiviral performance via the electrothermal effect on airborne viruses.

## 4. Conclusions

In summary, we newly presented the concept of on-demand antiviral ESP with electrothermal-based antiviral surfaces. We applied electrothermal-based antiviral surfaces to air-purifying applications and demonstrated that they are effective with regard to collecting airborne virus particles on collection plates in a two-stage ESP. With AC power, the MS2 bacteriophage and H1N1 viruses were completely deactivated after exposure to 50 °C for 30 min. This remarkable antiviral performance via the electrothermal effect indicates that using on-demand platforms for self-antiviral surfaces leads to immediate sterilization without generating secondary pollutants, thus effectively preventing the spread of infectious microorganisms in public places. Additionally, concerning energy consumption, the proposed system only required 5.16 W for disinfection, which is much lower than the energy required for other applications [38]. Moreover, the length of the proposed system was about 0.14 m, meaning that it occupies less space than other applications [38]. Furthermore, the ozone concentration, which is one of the main concerns of the proposed system, was significantly low (only about 10 ppb). Consequently, the proposed functional ESP has sufficient potential to be applied to indoor air-management applications. It can also be considered among other promising engineering applications due to its several advantages, such as high filtration and disinfection ability, low energy consumption, compact installation space, and minimum risk level to users. We believe that the results of this study provide a useful guideline for the design and realization of portable and wearable devices for antiviral air-purifying applications.

## Figures and Tables

**Figure 1 toxics-10-00601-f001:**
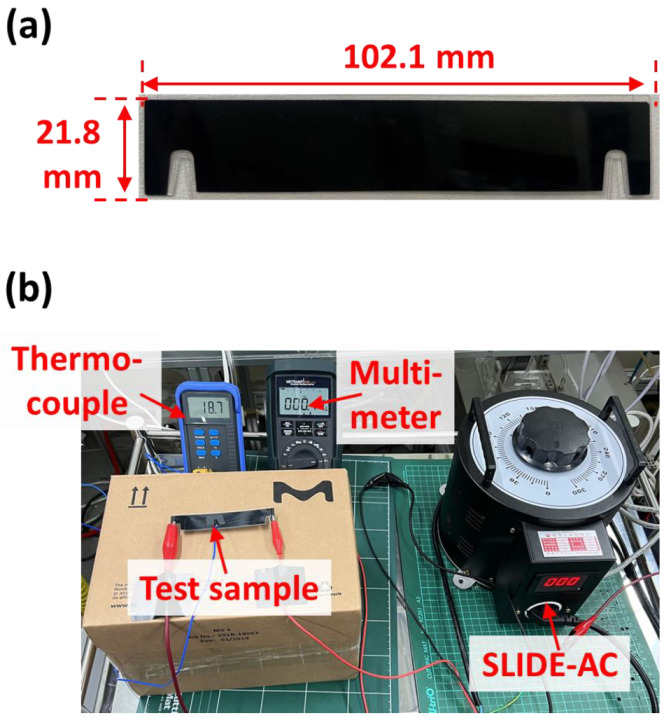
(**a**) Digital photo of the used carbon surface. (**b**) Experimental setup of the electrical resistance test for the characterization of electrothermal properties.

**Figure 2 toxics-10-00601-f002:**
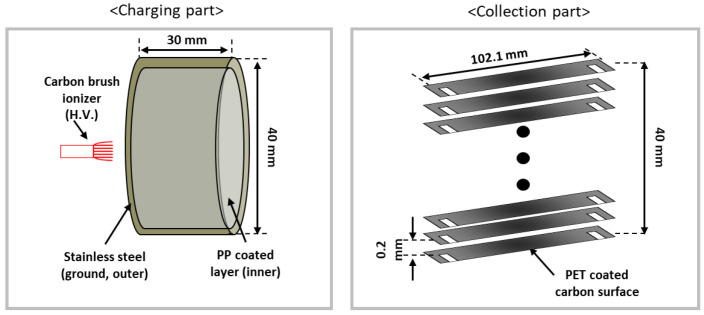
Schematic of the two-stage ESP. Detailed view of the charging part (**left**) and collection part with electrothermal carbon plates (**right**).

**Figure 3 toxics-10-00601-f003:**
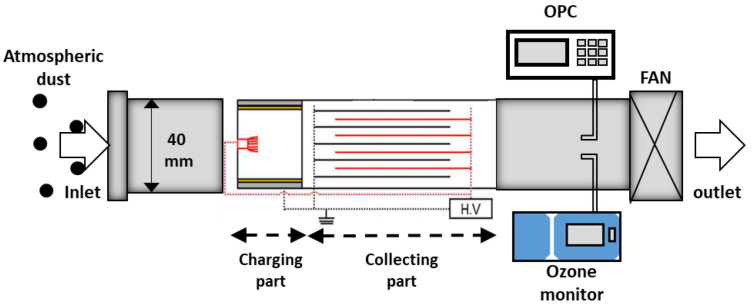
Experimental schematic for the single-pass PM removal efficiency test.

**Figure 4 toxics-10-00601-f004:**
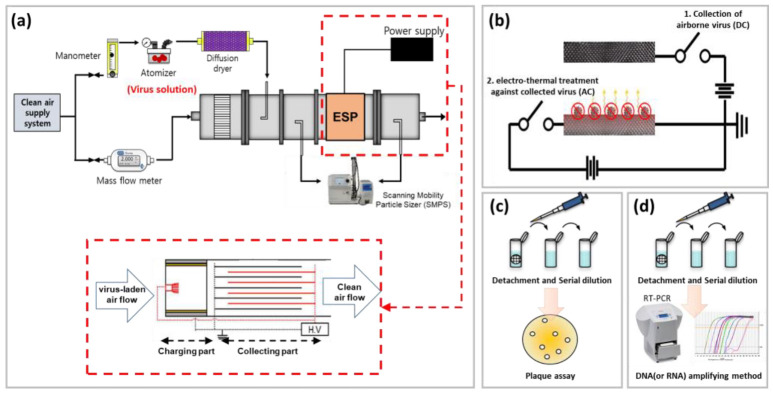
(**a**) Experimental schematic of the single-pass airborne virus removal efficiency test with ESP. (**b**) On-demand electrothermal antiviral process in ESP and evaluation of the antiviral efficiency against (**c**) MS2 bacteriophage and (**d**) H1N1 viruses.

**Figure 5 toxics-10-00601-f005:**
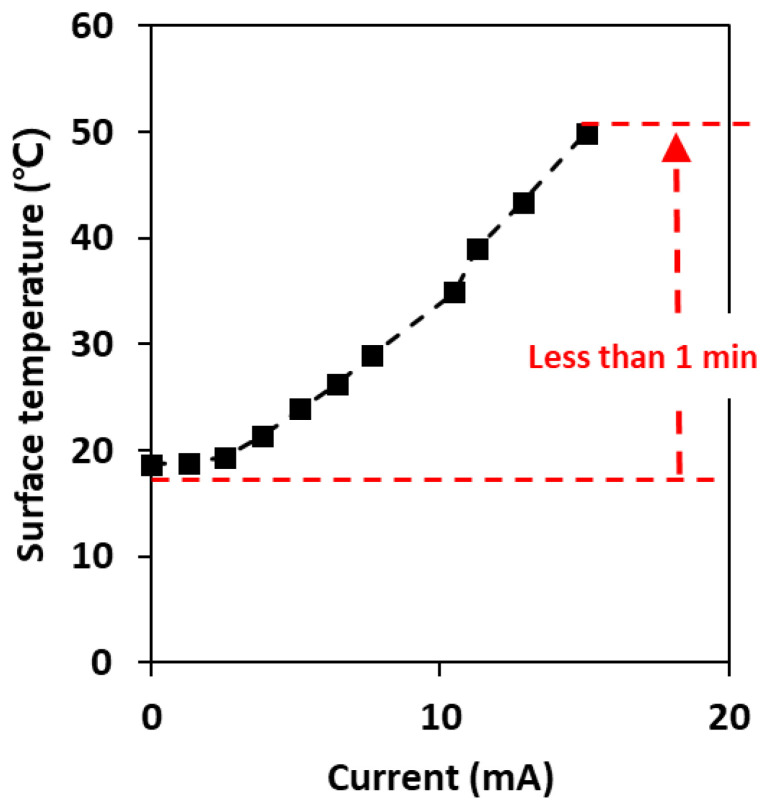
Surface temperature of the commercial carbon surface according to the applied current under AC conditions.

**Figure 6 toxics-10-00601-f006:**
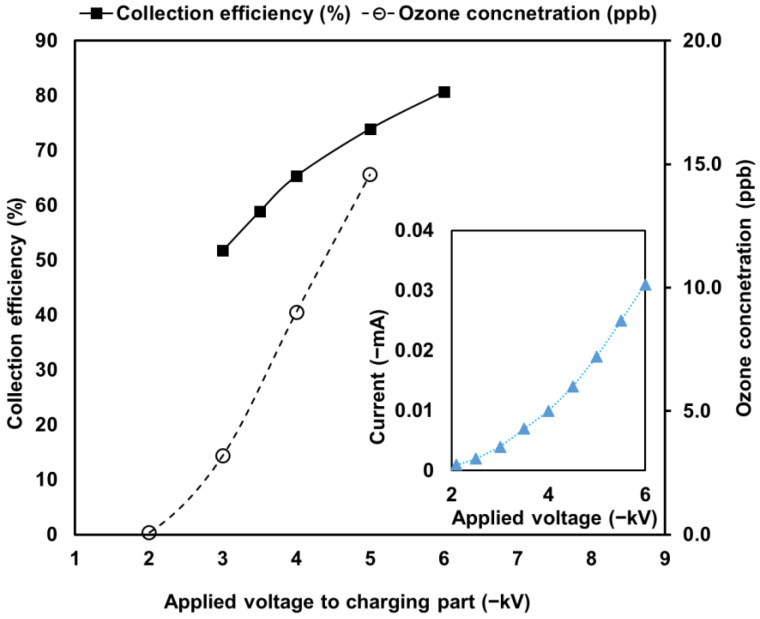
Corona discharge currents (inset), PM collection efficiencies, and ozone concentrations with the increase in the applied voltage to the charging part.

**Figure 7 toxics-10-00601-f007:**
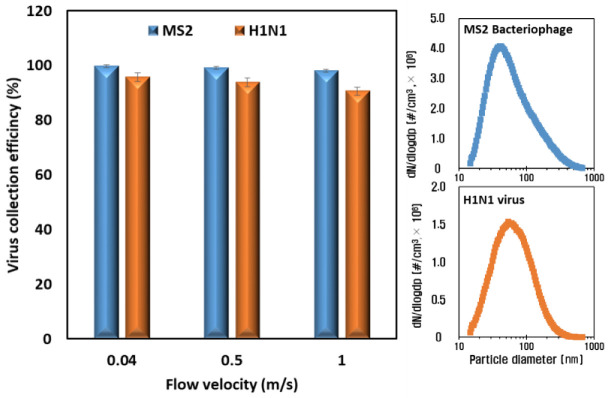
Collection efficiencies of airborne virus particles when increasing the face velocities (**left**) and size distributions of airborne virus particles (**right**).

**Figure 8 toxics-10-00601-f008:**
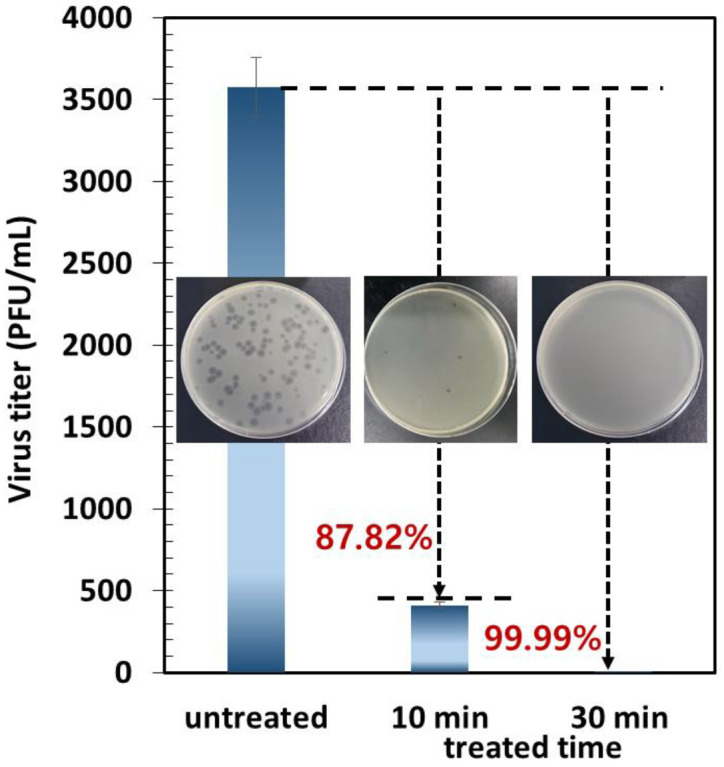
Electrothermal antiviral activity of the carbon surface against the MS2 bacteriophage via a plaque assay (inset photos: plaque assay results).

**Figure 9 toxics-10-00601-f009:**
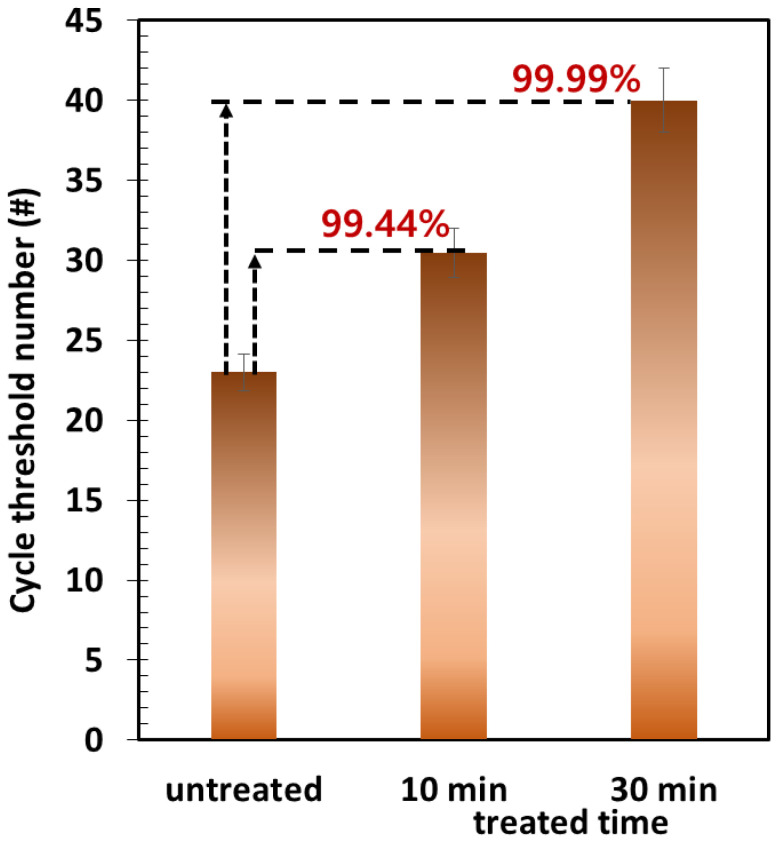
Electrothermal antiviral activity of the carbon surface against H1N1 viruses via a PCR system.

## Data Availability

Not applicable.

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
