# Peer review of "Development of On-Demand Antiviral Electrostatic Precipitators with Electrothermal-Based Antiviral Surfaces against Airborne Virus Particles"

_toxics, 2022, doi:10.3390/toxics10100601_

Round 1
Reviewer 1 Report
The purpose of this manuscript is to inactivate viruses on the collection electrode with electro-thermal. As the ozone concentration generated in the ESP and the collection efficiency for airborne virus are also indicated, the manuscript is suitable for MDPI Journal. However, I think the manuscript has a serious problem. Please revise the manuscript according to the comment before the publication.
1. Line 233
The authors described “…, the collection efficiencies against both virus species (MS2 and H1N1) are higher than 90 %, because the size distributions of the airborne viruses, mode diameters of which are 60 nm and 80 nm with MS2 and H1N1 virus particles in upstream of ESP respectively (Figure 7; right), have higher charging efficiencies than the 0.3 μm of atmospheric dust.”.
This explanation is not accurate. The amount of charge for the particle size of 60 nm and 80 nm are theoretically smaller than that of 0.3 um. This reason which the efficiencies are higher than 90% is considered that although the charge amount of the particle charged by corona discharge decreases with decreasing the particle size, the migration velocities for the particles of 60 nm and 80 nm are higher than that of 0.3 um due to decreasing the fluid drag force.
2. Please add the explanation why negative DC high voltage is used in this study. Because the ozone concentration generated in a positive corona discharge is lower than that in a negative discharge in generally.
3. Line 128
Please add the corona discharge current value. If possible, please add the graph showing the relationship between the applied voltage and the discharge current.
Reviewer 2 Report
This paper is well prepared and organized. The scientific contributions, innovative points and engineering application potential are satisfied. For the newly proposed air treatment system, the particle filtration efficiency, ozone removal effect and disinfection efficiency are well analyzed. Please analyze the energy consumption characteristics of this system. Please analyze the advantage of this system over the available systems, such as 10.1016/j.scs.2021.103226
Round 2
Reviewer 1 Report
I will recognize that this manuscript should be published.
However, I would like to make the following comments to the authors.
I consider that the high collection efficiency of particles smaller than 1 um is not because the main charging mechanism is diffusion, but because the migration velocity is proportional to the Cunningham coefficient, as shown in following reference (p.617).
Please use this as a reference.
1.) A. Mizuno, Electrostatic Precipitation, IEEE Transactions on Dielectrics and Electrical Insulation, Vol. 7, No. 5, October 2000, pp. 615-624, 2000